# Regulation for Promoting Sustainable, Fair and Circular Fashion

**Meital Peleg Mizrachi \* and Alon Tal \***

Department of Public Policy, Tel Aviv University, Tel Aviv-Yafo 69978, Israel
\* Correspondence: meitalpeleg@gmail.com (M.P.M.); alontal@tau.ac.il or alontal48@gmail.com (A.T.)

**Abstract:** Over the past few decades, the production and consumption of clothing has increased exponentially, leading to a dramatic increase in the negative environmental consequences produced by the fashion industry. Given the rising pace of global warming and the rising concern about the fashion industry's contribution to the climate crisis and its exploitative social dimensions, decision makers, politicians and government officials have begun to promote sustainable fashion through public policy. This article reviews the main barriers facing a circular economy in general and the fashion industry in particular. It considers nascent regulations emerging throughout the world in the field of sustainable fashion, applying circular economic principles to the fashion industry. Four categories of policy proposals are evaluated: Command and Control Interventions, Educational Initiatives, Incentives and Certification, grading them according to criteria of effectiveness, sustainability, feasibility, equity and compliance. The ranking process was based on an elicitation of expert judgement among a panel with expertise in the areas of sustainable fashion from the business sector, academia and civil society. Findings suggest that while some policy options appear to meet all criteria successfully, when selecting an optimal strategy for promoting sustainable fashion, there are frequently trade-offs between different alternatives.

**Keywords:** sustainable fashion; circular economy; sustainability; regulation; public policy; economic barriers; policy proposals

## 1. Introduction

*Environmental and Social Aspects of the Fashion Industry*

Over the past forty years, the amount of clothing produced in the world has increased by 400 percent [1]. At the same time, household spending on clothing has decreased by a few dozen percent [2,3], along with the average number of times a garment is worn. These trends reflect the penetration of the fast fashion model, improvement in technological capabilities and extremely rapid turnover. At first glance, such changes might be interpreted positively—trends that serve to increase consumer purchasing power. The process is often referred to as the "democratization of fashion" [4]. However, the dramatic transformation has far-reaching, adverse environmental and social consequences.

The fashion industry is responsible each year for emitting 1.2 billion tons of greenhouse gases; releasing half a million tonnes of microplastics into the sea [5]; using 132 million tonnes of coal [6], 9000 million cubic meters of water [6] and a quarter of the world's toxic chemicals [7]. It also creates billions of tonnes of non-recyclable textile waste [5].

The extent of the destruction caused by the fashion industry is expected to continue to grow; by 2030 the fashion industry will use 35% more fiber-growing space, requiring an additional 115,000 square kilometers of land [5]. Textile waste is a primary contributor to the rapid depletion of land-fill capacity. From the 1960s to 2018, there was an 800% increase in the amount of disposed textiles buried in landfills [6]. In fact, despite rosier perceptions, only one percent of used clothing is recycled [8]. Emissions, water contamination and soil destruction during the production process, along with the environmental consequences

caused by burying and burning clothes at the end of use will primarily take place in the developing countries of Southeast Asia and Africa. Most of these clothes will be purchased in Western countries.

The fashion industry's balance sheet also reflects additional examples of environmental injustice. Western consumers hold a staggering amount of clothes in their closets. The glut leads to 85% of the clothes purchased in the USA being thrown away in less than a year from the moment of purchase [9]. Some 98% of the workers in the fashion industry do not earn salaries sufficient to feed their families [10] and 64% of the women employed in the textile factories suffer from physical and verbal abuse on a regular basis [11]. These figures suggest that, in addition to being the second most polluting industry, the fashion industry is also the second most exploitative one in the world (second only to the cell phone industry).

This article seeks to address the problem of externalities and inequities created by the fashion industry through the evaluation of regulatory strategies designed to advance sustainable, fair and circular fashion. It reviews existing regulation, the barriers to advancing a circular economy in general, along with the obstacles to a circular economy in the field of fashion in particular.

The article begins with a brief description of the principles of circular production, their nascent regulatory expression, challenges and how they might inform and improve the global fashion industry's practices. The next section reviews new government programs in Europe and beyond for addressing the sustainability challenges of the fashion industry. It suggests that there are growing opportunities for the increased regulation of environmentally destructive consumption and interventions in the field of the fashion industry in particular. The article then considers a suite of possible public policies at the national level to engender progress in encouraging sustainable fashion consumption and assesses them according to the criteria of effectiveness, sustainability, feasibility, equity and compliance. Four categories of policy proposals are evaluated: Command and Control Interventions, Educational Initiatives, Incentives and Certification.

The analysis suggests that while some types of interventions may appear to be easier to implement and enjoy ranking, there are still trade-offs between competing criteria. For instance, a government intervention may lead to improved environmental performance but may do little to address problems of social equity or political feasibility. Nonetheless, as regulatory programs to address the externalities of the fashion industry emerge, the need for a systematic assessment of their relative advantages and disadvantages is of increasing importance.

## 2. Background: Regulation Promoting the New Circular Economy

In July 2021 a metaphorical earthquake occurred in the European economy. The EU began imposing a border tax under which Value Added Tax (hereinafter: VAT) will be levied on personal imports from abroad without limitation [12]. Under the new regulations, present VAT exemptions for the importation of small consignments up to a value of 22 EUR will be cancelled. This means all goods imported into the EU will now be subject to VAT. The rationale for these changes addresses the VAT regimes for distance sales of goods and the importation of low value consignments which are not produced within the EU.

The VAT exemption granted prior to July 2021 created several problems: the primary one involved imbalanced competition between local businesses in Europe and manufacturers and distributors abroad. The exact same product sold at a local and foreign site would be priced differently, with the foreign site less expensive—solely due to the tax gap. In other words, the tax system incentivizes consumers to purchase internationally, which increases transportation expenditures and associated environmental impacts. Elimination of the exemption reduces the possibility of VAT fraud and increases VAT revenues for EU member states by as much as 7 billion euros [13].

The new provision applies to everyone: suppliers and deemed suppliers, EU and non-EU sellers, consumers in the EU, and third territories and third countries. Those who

are likely to be most affected by the new law, however, are large online trading companies such as Amazon and eBay, fast fashion companies that have shifted most of their business to online trading arenas, and fashion corporations that rely solely on online sales, such as ASOS and their East Asian suppliers.

This new taxation policy can potentially contribute to reducing the negative environmental consequences of consumption in general, and fashion consumption in particular. It encourages local consumption and production; helps reduce environmental transportation costs; facilitates the regulation and oversight of employment conditions in supply chains, as well as the environmental performance of manufacturers; and significantly reduces the opportunity to purchase inordinately cheap products, [12,13] which may, as a result, lead to a reduction in overall consumption.

The associated environmental benefits are all highly germane to the fashion industry, which is now considered to be the second most polluting industry in the world (energy retains the dubious honor of number one polluter), partly due to the rise and domination of online commerce.

In this context, it is important to remember that although the new EU border taxation policy constitutes a modest contribution to the above dynamics, in practice, it constitutes the tip of the proverbial iceberg of change. The damages caused by the fashion industry are so extensive that in order to meaningfully influence the fashion consumer, at the end of the production chain, a comprehensive economic strategy is required [14]. Policies need to address all stages of production and actively encourage the formation of new and sustainable business models in fashion, offering radical substitutes that pose an alternative to the prevailing, linear business model.

It is important to emphasize that while the most significant harm to natural resources and workers' rights takes place during the industrial production process, the forces that actually drive destructive activity are consumption and accelerated economic development. These include consumer preferences. In fact, about two-thirds of global greenhouse gas emissions are directly and indirectly related to individual consumption [15]. Accordingly, it is important to encourage environmentally and socially responsible growth and development through sustainable consumption. Knowing that clothing constitutes the single most common consumer product in households, it can be argued that sustainable fashion has a critical role in sustainable lifestyles and individual greenhouse gas reductions.

### 2.1. The Circular Economy

Promotion of a circular economy, combined with economic regulation, can be a very important tool for influencing consumers to prefer sustainable fashion over fast fashion. The circular economy offers an alternative economic model to the linear economy, promoted by the liberal worldview since the beginning of the Industrial Revolution [16]. A circular economy seeks to sever the link between growth and economic activity and the consumption of virgin and perishable natural resources [17]. It proposes an alternative model that supports sustainable development, social justice and economic well-being [16,17]. A circular economy strives to eliminate the waste outside the system, by relying on new business models (such as collaboration or use without ownership) [18]; designing products for long-term use with minimal residuals in production [19], as well as designing products and materials in a way that allows them to be repaired, or at least easily recycled [16]. In short, it designs a production process that reflects the circular patterns found in natural systems [20], with minimal depreciation and emissions along with maximum reuse of materials.

Wise consumption means minimizing unnecessary purchases, preferring products with circular characteristics and choosing cooperative models of consumption that prevent the storage and disposal of finished products and also contributes to the circular economy [21]. Finally, reusing, repairing and recycling addresses the by-products of production and consumption processes, enabling the creation of secondary raw materials, capable, in some cases, of replacing virgin raw materials mined from nature [22]. This model strives

for efficiency and is supported by technological innovation that contributes to business competitiveness, job creation and, at the same time, environmental protection. Hence, creating a circular economy requires substantial and profound changes in existing production paradigms, as well as a more equitable sharing of environmental burdens in production and consumption. Expediting such changes requires a more supportive infrastructure of public policies and regulation [23].

A circular economy approach in the field of fashion addresses design strategies which include reduced utilization of virgin raw materials, efficiency, recycling, reuse, and re-manufacturing, new business thinking, avoiding textile waste, slowing down consumption. It also embraces new business strategies which include renting, sharing, swapping, and borrowing, while at the same time increasing sustainable fashion consumption (See Figure 1) [24].

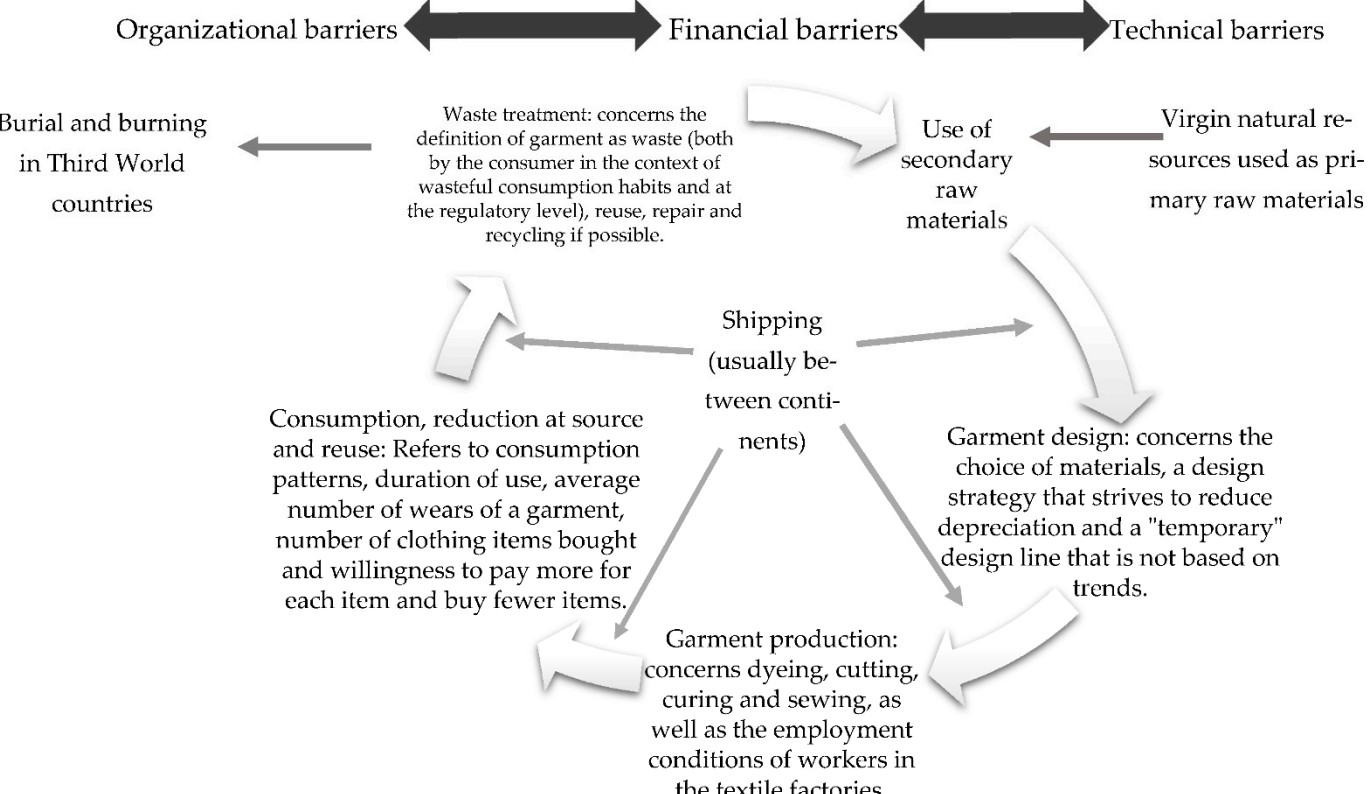

**Figure 1.** An abstract model of a circular economy, including barriers, in the field of fashion.

It should be mentioned that the definition of sustainable fashion is not uniform and can even be controversial [25] Buying clothes from sustainable materials contradicts the broader objective of reducing purchases of new products. Buying from local designers can conflict with the use of existing clothes or buying second-hand clothes. It also makes it significantly more difficult to maintain transparency in the production process. Common to all approaches, however, is the desire to create an alternative to the fast fashion industry and reduce the environmental and social impacts of the fashion industry [26].

### 2.2. Dilemmas Pertaining to Sustainable Fashion and Its Expression in the Business World

We find it appropriate to point this out, as there are also voices in the field of sustainability who would argue, with some justification, that reducing and stopping fashion consumption, which unlike food is not a life-sustaining product, is the most environmental solution to the problems posed by today's fashion industry [27,28]. It is, however, important to emphasize the distinction between fast fashion and sustainable fashion. In our view, increasing the market share of sustainable fashion should lead to a reduction in the

consumption of fast fashion as well as fashion in general, in a way that still allows the world to enjoy the creativity and joy of fashion without excessive environmental and social costs. At the very least, the transition from fast fashion to sustainable fashion, is an important intermediate step on the way to reducing overall consumption. In this spirit, in July 2021 seventeen leading UK fashion companies, representing 50 per cent of the country's clothing and textile sales, signed a voluntary agreement: "*Textiles 2030*", to reduce their greenhouse gas emissions by 50 percent, their water consumption by 30 percent, and integrate new approaches of circular economy into their business model by 2030 [29].

The agreement includes a clear plan for achieving the goals over the next decade and a detailed blueprint to guide the fashion industry should it seek to become circular. Among other measures, the report recommends producing clothes that will last longer and that are easier to recycle at the end of their lives [29]. The proposed production systems would not produce waste and rely primarily on recycled raw materials. The *Textiles 2030* report is the product of a collaboration between government, business, community organizations and NGOs. It was written with the aim of encouraging governments, businesses, financiers, investors and non-profit organizations to take broader action that includes "radical and significant changes".

Along with its recommendations, the report also emphasizes the inherent need for supportive regulation. In fact, according to the report, without supportive economic regulation, it will not be possible to reach the report's stated objectives, as well as global emission reduction targets [29].

This conclusion is consistent with the UN *Fashion Industry Charter for Climate Action*. The charter laid out new goals for its 130 signatories. It was launched during the second week of the UN Climate Conference COP26, under the leadership of Stella McCartney and the United Nations. Under the Charter, signatories must submit action plans within the next 12 months detailing their pathway to achieve the Charter's upgraded goals. Under the updated charter, signatories must pledge to reach net-zero emissions by no later than 2050 and either halve their emissions by the end of the decade or set science-based targets within the next 24 months [14].

About a week before the launch of the Charter, the *Nonprofit Textile Exchange* launched a call backed by more than fifty fashion and textile companies. The exchange includes such industry leaders as Gucci-owner Kering, Gap and H&M Group. In addition to its declared "burning need" for regulation and environmental policy in the field of fashion [14], the Charter called for changes to trade policy that would encourage the use of so-called preferred materials, such as organic cotton or recycled fibers:

"It's time to accelerate progress", said Stefan Seidel, Puma's head of corporate sustainability and chair of the Fashion Charter steering committee. "Now with the new ambition level, we're all clear on where we need to go" [14].

### 2.3. Barriers to a Circular Economy

In the last decade, many examples of business models that combine a circular economy, such as lease, pay-per-use and take-back schemes, have become widespread. Several evaluations of present production systems indicate that the current institutional economic frameworks hinder an expeditious transition to a circular economy [30]. These studies suggest that in order to achieve the United Nations Sustainable Development Goals (SDGs), we will have to change the rules that currently govern our economic system in close cooperation with government institutions [16,22,30]. In this context, it is worth mentioning that the UN has defined the fashion industry as being "of high relevance to integrating the targets of the seventeen Sustainable Development goals". [31].

Accordingly, in order to promote a transition to a circular economy and to achieve the SDGs, the main barriers facing a circular economy in general, and to sustainable fashion in particular, must be mapped.

A circular economy requires changes on three fronts: technical, financial and organizational [30]. The technical "circularity" aspect refers to thinking about how we manufacture

the products. Production of products within a circular economy refers to the recycling, renewal, repair and extension of product life. In this context, most of the barriers merely interfere with firm capacity—that is, they require new technological-industrial developments. However, initially, changes must also take place in the visions and decisions of the relevant manufacturers. Such factors are greatly influenced by the two other forces: the financial dynamics and the organizational structures [32–34].

As to the financial aspects, from an economic point of view, four main barriers have been identified that directly relate to the revenue model of businesses:

1. Common depreciation standards in accountancy incentivize organizations to regard a product's value as declining rapidly toward EUR 0. This stimulates a take-make-waste model [21]. Present rules serve to increase the tax benefits that producers can obtain. Rapid depreciation rates lower the perceived market value of used products, which constitutes a barrier to the development of a circular economy for which used product value is a necessary precondition. Furthermore, depreciation standards also limit the maximum length of rental, lease or pay-per-use periods [33].

2. VAT favors the traditional sales model over the more circular rent-purchase model. In a linear economy, most producer–user relationships rely on traditional sales relationships that involve paying the full value of the product at the time of sale. As a result, manufacturers are required to pay VAT on the income earned at the time of sale. Producers, however, operating rent-purchase relationships with customers, still need to pay VAT on all projected revenues obtained during the rental period, as rent-purchase is seen as the deferred supply of goods [30,35]. This results in negative business dynamics. If entire businesses are built on this relationship, upfront costs will have negative impacts on liquidity and business viability.

3. VAT currently does not favor used products and materials over new ones. In addition to the barriers described above, concerning VAT, in B2C transactions, new, second-hand and recycled products are all taxed equally. This means that for all products that are not new, the sales tax is paid twice or even more by users—once at every transaction of the product. This reduces business competitiveness, especially considering the extra costs incurred in creating take-back systems [36].

4. Common financial assessment practices favor linear economic models over circular economic models. The less-certain financial nature of circular revenue models (CRMs) makes them riskier from a traditional financial risk assessment's point of view. CRMs are characterized by recurring periodic revenue streams and therefore longer payback periods. They also represent a value shift from assets to contracts [21,34,37].

Traditional risk assessments of CRMs tend to address them as posing greater financial risks than traditional revenue models. There are several reasons for this: balance sheet extension due to producer ownership; uncertainty about incoming cash, especially with B2C relationships (debtor risk); uncertainty regarding contract length due to increased contract flexibility; and contract financing, which is seen as riskier than asset financing [21,30,37]. On the other hand, CRMs offer a number of advantages, which are rarely accounted for in traditional financial risk management. Beyond the obvious environmental benefits, there are also longer product lifetimes; higher residual values; more resilience towards future resource scarcity; and opportunities to expand to new (second-hand) markets.

These barriers ensure that linear income models are maintained as a status quo and continue to serve as the default paradigm for business models. Other finance barriers relate to accountancy standards for leasing (i.e., IFRS). These include easy access to finance for governments, waste legislation and taxing labor as opposed to taxing resources. Nonetheless, with simple changes in economic regulation, these four main barriers can be overcome with relative ease, generating disparate benefits in the field of CRM [30].

In addition, circular reuse must be organized "up front", in order to ensure the product's circularity throughout the value chain. Otherwise, a product, in theory, can be designed and produced according to the "technical" principles of circularity. However, there is no guarantee that in reality the product (and its components/raw materials) will indeed

be used in a circular manner, either by manufacturers or by consumers. As Mark Hogarth, creative director at Harris Tweed Hebrides, explains: "Clothes can be very sustainable, but what will determine their sustainability is how they are used by consumers" [38]. For example: a Cradle-to-Cradle pair of jeans is circular in theory. However, it will only contribute to a circular economy if we prevent it from ending up in a landfill [30,38]. Therefore, cooperation between the chain partners, including business owners, consumers and policy makers, is essential for the transition to a circular model.

An additional significant organizational barrier is that the suppliers are not sufficiently stimulated to redesign their products to facilitate value retention, nor is value retention actually being incentivized. Value retention means designing a product so that it can last for as many years as possible, both in terms of quality and relevance. With regard to fashion, it means the production of high-quality clothing that are resistant to the dangers of time and laundry. The design of high-quality clothing also contrasts with the rapid pace, at the heart of today's fast fashion model.

Indeed, there is a fundamental conflict with the defining ethos that informs the fashion industry. The concept of value retention conflicts with the need to promote the purchase of new products, even if they are clothes in the spirit of the circular economy, for example, made from recycled materials. Hence, frequently, model programs designed to promote a circular economy can lead to recycling. While preferable to disposal, the programs usually lower the value of the material, creating many environmental impacts and making it technically almost impossible for the fashion industry to pursue maintenance, reuse or remanufacturing initiatives. Yet, we argue that the approach applied in the circular economy can be significantly less expensive. Given the state-of-the-art technology for textile recycling, today, more than ever, a circular approach is achievable.

Another organizational barrier concerns the willingness of consumers to purchase products. Here, utility models of a circular economy, as opposed to a linear economy are relevant. Many consumers are not aware of the environmental benefits of a circular economy and the value of using recycled materials in consumer products. In the absence of a significant incentive, economic or social, they a priori, consider circular economy products in all forms, to be inferior and as a result, avoid purchasing them [39,40].

*2.4. Barriers to a Circular Economy in Fashion*

Many of the barriers facing a circular economy described above are manifested in the fashion industry. In just one of many examples, the economic barrier of VAT, which does not favor rent-purchase relationships, negatively affects the dynamics of MUD Jeans: MUD Jeans rents jeans for a fixed monthly fee through its Lease-a-Jeans concept [30,41]. After one year, users can decide to keep the jeans, and turn the rental transaction into a purchase transaction, or swap them for a new pair. Thus, MUD Jeans encourages its customers to use jeans for a longer period, while at the same time, recycling the returned jeans as new jeans.

Notwithstanding the obvious environmental benefits of the circular economic model of MUD jeans upon first payment of the jeans, MUD Jeans only obtain 1/12 of the alternative sales revenues. However, retailers need to pay the full VAT on all revenues obtained during the subscription period. Thus, current VAT rules have a negative effect on the revenue model of MUD Jeans, limiting its scalability.

One telling instance of this distortion in taxation can be seen in the case of ThredUP. ThredUP is a one of the largest online thrift stores in the US, offering second-hand items from a variety of brands such as H&M, Forever 21, Gap, and Banana Republic, alongside premium brands from top designers [42]. TherdUP also allows customers to sell clothes in which they have lost interest in exchange for the transportation and handling costs. Despite the many environmental benefits of this transaction, from the point of view of many consumers, the second-hand clothes it offers are an inferior product, when compared to the same apparel sold in its original form in retail stores.

Unlike conventional clothing stores, ThredUP sells clothes after they have already undergone a number of uses and washes. This can detract from their quality. In addition,

each garment appears once and in one size, so the range of sizes is significantly reduced. As a result, many consumers expect to find the clothes at a lower price than the price at which they were first sold by producers. Additionally, indeed, on the ThredUP website, they guarantee a price lower than the original price by up to 90% [42]. Despite the significant reduction in profit, ThredUP is taxed no differently than a fast fashion corporation. This individual case faithfully represents the fiscal reality of most of second-hand shops, which constitute an important part of a circular economy in the field of fashion. It is particularly true, given the astonishing fact that there are already enough clothes in the world to dress humans for the next fifty years [6].

Another critical technical barrier facing sustainable fashion concerns the recycling of clothing. Worldwide, 130 billion garments are produced annually, of which 80 billion are sold, yet less than a single percent are recycled [7,8,43]. Low recycling rates can be attributed to a number of factors: most of the clothes produced today are made of mixed materials (cotton, polyester, Lycra—Spandex, etc.) which are essentially non-recyclable; textile recycling programs requiring a substantial human workforce to sort the clothes according to the compound of the fabric and color, as a preliminary stage for recycling; present recycling technology today remains significantly more expensive than burning or burying clothes. When these technological factors combine with the absence of sufficient regulation prohibiting burning and burying clothes, most clothing manufacturers simply take the "path of least resistance" and choose not to recycle [43].

It should be mentioned that the environmental impact of low-cost garments typically is not felt in the location where they are purchased but rather primarily affect the locations where they are produced, buried or burnt [44]. This geographic disconnect constitutes an additional barrier, allowing low-cost, garment manufacturers to overlook, or intentionally ignore the environmental costs in the final price of the product. Basic regulatory axioms involving the "polluter pays principle" as well as policy tools designed to internalize externalities have not yet been applied to this globalized market.

Accordingly, a significant organizational barrier facing textile recycling involves legislation [45–47]. Neither European nor American environmental laws target textile waste management. Regulations vary between countries and there is no legal mechanism that mandates or regulates the collection of textiles. A collection requirement for post-consumer textiles offers the potential of establishing reverse supply chains and rearranging industrial operations, as well as influencing the disposal practices of consumers [48,49].

Accordingly, one of the most significant, prevailing organizational barriers to a sustainable fashion industry involves consumers. A circular economy is not solely about technical processes, recyclable materials or recovery in a transition to renewables" [50]. Attention in the discourse surrounding a circular fashion economy tends to focus on manufacturers, with reference to production methods and new business models. When it comes to fashion, it is simply not possible to promote a circular economy without the thoughtful integration of consumers. As the Swiss, sustainable production expert, Walter Stahel posits: "the optimization of use. Or utilization of manufactured objects, is at the core of the circular economy" [24].

It is therefore important to note that most fashion consumers not only fail to optimally utilize their clothing, but actually exhibit wasteful patterns of usage. On average, 80% of the time, fashion consumers use 20% of their clothes [7]. Moreover, 21% of the clothes in the closets of developed countries will never be worn [40]. An estimated 85% of clothes purchased, whether in a circular economy or in a fast fashion outlet, will be disposed of in less than a year [9,51]. Hence, the effectiveness of efforts to promote sustainable fashion, which do not require consumers to make radical changes in their consumption practices are likely to be limited. Circular economy programs that only include alternative manufacturing methods and focus solely on manufacturers are not sufficiently comprehensive to solve the fundamental problems associated with the fashion industry.

Another lacuna in present circular economy methods involving the fashion industry is the narrow focus on the environmental aspects of production at the expense of adequate

consideration of social aspects. As mentioned, the fashion industry is not only one of the most polluting industries in the world, but also one of the most exploitative. The collapse of the Rana Plaza textile factory in 2013 in Dhaka, Bangladesh, in which more than 1000 workers were killed and another 3000 injured, brought the issue of workers' rights in the textile factories to international attention, due to the extraordinary scale of the disaster [52].

Tragically, work accidents in the fashion industry remain very common: Since January 2021, there have been thirty-six incidents and fatal accidents as a result of faulty safety measures in textile factories, leading to the deaths of 109 workers and 153 serious injuries in Pakistan, India, Egypt, Morocco, China, and Cambodia [53]. Furthermore, accidents leading to occupational deaths and injuries typically frequently are not reported at all. It can therefore be assumed with high probability that the number of preventable deaths from clothing manufacturing is significantly greater than present estimates [53].

Therefore, it is important to remember that sustainability, in its broad conception, relates to the tripartite relationship between humans, environment and economy [20]. It is not possible to chart a sustainable course for the fashion industry without addressing the conundrum of employee protection and occupational health and affording it the same importance as environmental protection. A broader concept is needed, one which sees a circular economy as a means of promoting sustainable fashion and not merely a goal.

Finally, it is clear that voluntary regulation on the part of fashion manufacturers, despite its symbolic significance, is not sufficient to address the many problems posed by the fashion industry. In order to realize the SDG and UNFCCC emission reduction targets, as well as ensure the safety of textile workers around the world and achieve a radical change in consumers' fashion consumption patterns, local, national and in particular, Transboundary regulation is essential. The following section reviews existing regulations and proposes possible regulatory strategies for promoting sustainable fashion, with an emphasis on establishing a circular economy.

### 3. Policy and Regulatory Programs in the Field of Fashion and Circular Economy

Over the past decade many cities, countries and public bodies have put forward a myriad solutions in an attempt to address the environmental and social implications of the fashion industry. In June 2015, in the wake of the Rana Plaza disaster, the G7 Leaders' Declaration welcomed international efforts to promulgate industry-wide due diligence standards in the textile and ready-made garment sector [54].

In 2019, the United Nations launched *The United Nations Alliance for Sustainable Fashion*, with the aim of contributing to sustainable development goals, through coordinated action in the field of fashion. Many municipal governments were already on board: In 2019 the Stockholm Municipality canceled *Fashion Week* in order to find more environmentally friendly alternatives [55]. At the same time, the Paris Municipality announced that it aspired to become a sustainable fashion capital by 2024. Below is a summary of some of the major national initiatives promoting sustainable fashion around the world. The review is not exhaustive and does not seek to present every sustainable fashion initiative. Nonetheless, it offers an indication of the breadth and diversity of recent initiatives in this emerging field. Figure 1 and Table 1 map the initiatives and regulations according to the stages of production in a circular economy, as well as the degree of effectiveness of each initiative, for each of the steps.

**Table 1.** Mapping existing initiatives and regulations in the field of sustainable fashion, according to stages of product in a circular economy. black = likely to have a large effect, dark gray = likely to have a moderate effect, Light gray = not likely to have any effect at all.

| | Waste Treatment | Reduction within the Source (Waste Minimization) | Fashion Consumption | Garment Production | Garment Design | Use of Secondary Raw Materials |
|---|---|---|---|---|---|---|
| Due Diligence Guidance for Responsible Supply Chains in the Garment and Footwear Sector. 2017 | light gray | light gray | dark gray | black | dark gray | light gray |
| The Australian Modern Slavery Law 2018 | light gray | light gray | dark gray | dark gray | light gray | light gray |
| The French law banning the destruction of unsold clothing 2019 | black | dark gray | dark gray | light gray | dark gray | dark gray |
| New York State regulation requires businesses with more than 10% of its waste comprised of textiles to send residuals for recycling 2021 | black | dark gray | light gray | dark gray | dark gray | black |
| The sb62 bill 2021 | light gray | dark gray | light gray | black | light gray | light gray |

In 2017 the OECD published the *Due Diligence Guidance for Responsible Supply Chains in the Garment and Footwear Sector*. The guide aims to help enterprises implement the "due diligence" recommendations contained in the OECD *Guidelines for Multinational Enterprises* along the garment and footwear supply chain. Its stated objective is to avoid and mitigate the potential negative impacts of its activities and supply chains, and to support Implementation of the International Labor Organization's (ILO) Tripartite Declaration of Principles Concerning Multinational Enterprises and Social Policy [56]. The guidelines were published under the premise that the common aim of governments is to promote responsible business conduct [57]. The Guidance, which was developed through a multi-stakeholder process, has been approved by forty-eight governments and endorsed by all forty-seven OECD countries. This constitutes about 72% of the clothing importers in the global industry, business, trade unions and civil society [58].

In December 2018, the "Modern Slavery Law" came into force in Australia. According to the law, companies with revenues above 100 million dollars (Australian) are required to publish an annual *Modern Slavery Statement*, reporting on all potential modern slavery risks and practices in their operations and supply chains [59]. These modern slavery statements are required to address an entity's structure, operations and supply chains, any actions taken to address modern slavery risks and the effectiveness of such actions. All statements are made publicly available in a central government-run repository to foster public oversight. Penalties are to be applied for non-compliance with the reporting requirement. The aim of the law is to increase transparency around modern slavery and human trafficking in supply chains for consumers and investors and improve workplace anti-slavery practices by holding businesses accountable [60].

In France, as of 2019, fashion sellers and retailers are prohibited from throwing away discarding or incinerating unsold clothes. At the same time, they are required to collect unsold clothing and prohibited from disposing unwanted clothing in conventional landfills.

The law requires clothing companies to donate the surplus clothing and unsold clothing, thus forcing them to undertake substantial responsibility for production quantities as well as logistical measures [61,62]. The law was later integrated into the national *Food Act* [63], which previously banned supermarkets from throwing away unsold food. The program was deemed an unprecedented success, as part of France's new *Circular Economy Roadmap*, designed to eliminate product waste [63].

In New York State, in February 2021 a new regulation was enacted, extending manufacturers' warranties. The regulation also includes reference to the textile sector and stipulates that any business with more than 10% of its waste comprised of textiles is obligated to send residuals for recycling. The fines set in the regulation are low and only amount to several hundred dollars, a negligible amount for medium and large manufacturers. Yet, more important than the fines, which can be seen largely as a policy "nudge", the operational requirements promulgated push businesses to operate extensively in the recycling field and adopt circular economy practices. These dynamics are expected ultimately to help extend garment life somewhat. In the deeper sense, this regulation not only expands manufacturer liability, but also push markets to think of non-linearly and consider wasteful business models [64].

In September 2021, the sb62 bill was passed in the California Legislature, addressing workers' rights in the textile industry, within western American states. The law prohibits payment by output, payment below the threshold allowed in the wage laws and exploitation resulting from lack of follow-up. The bill requires apparel factories to pay garment workers an hourly wage and bans the long-standing piece-rate system: 5 cents to sew a side seam, for instance, or 10 cents, to sew a neck. This often adds up to less than USD 6 an hour, while allowing employers to still offer productivity-based incentives to workers. The new California law also allows workers to demand the return of stolen wages from large fashion brands and retailers. The meaning of the initiative is essentially declarative: it calls for substantive and legal responsibility from major fashion brands and prevents them from relying on outsourcing with its associated labor exploitation and environmental damage [65] (for a summary of the initiatives and regulations according to stages of product recycling in a circular economy, see Table 1).

The Pareto principle, which was formulated by French economist Wilfredo Pareto in the nineteenth century, states that for many outcomes, roughly 80% of consequences come from 20% of causes. In this spirit the policy proposals detailed below address a wide range of problems in the fashion industry, with no single proposal likely to solve all the problems created by the industry. Nonetheless, the analysis seeks to identify the 20% of change required to affect 80% of the associated challenges. For a spatial presentation of the proposed alternatives according to the stages of the product cycle in a circular economy, along with the degree of effectiveness of each alternative for each of the steps see Figure 1 and Table 2.

The sustainable fashion policy alternatives assessed can be divided into 4 categories: Command and Control, Incentives, Educational and Certification. The common denominator of these disparate interventions is the encouraging of sustainable fashion and a circular economy in the field of fashion. The policy proposals will be systematically examined on the basis of four criteria: effectiveness, equity, sustainability and political feasibility, as explained above.

**Table 2.** Mapping proposed alternatives, according to the stages of the product cycle in a circular economy. black = likely to have a large effect, dark gray = likely to have a moderate effect, light gray = not likely to have any effect at all.

| | | Waste Treatment | Reduction within the Source (Waste Minimisatio) | Fashion Consumption | Garment Production | Garment Design | Use of Secondary Raw Materials |
|---|---|---|---|---|---|---|---|
| Command and Control | Household Waste Fees | black | black | black | dark gray | light gray | black |
| | Legal liability for violation of garment workers' rights | light gray | light gray | light gray | black | light gray | light gray |
| | Advertising Standards for the Apparel Industry | light gray | dark gray | black | black | light gray | light gray |
| Educational | Educational campaigns to reduce consumption | light gray | black | black | dark gray | light gray | light gray |
| Incentives | Carbon taxes on clothing | black | black | light gray | black | black | black |
| | Incentives and investments on favorable terms to businesses offering sustainable fashion services along with reduction of regulation and taxation for thrift stores | dark gray | black | dark gray | black | black | black |
| Certification | Standardization or certification of sustainable fashion | dark gray | black | black | black | black | dark gray |

### 3.1. Command and Control Interventions

3.1.1. Household Waste Fees

One regulatory approach in this field involves the imposition of a waste removal fee along with reforming the definitions of solid waste laws in the field of fashion and textiles. The definition adopted by the EU's current *Waste Framework Directive* (WFD in 2008) [66] for waste dictates that a substance owner's behavior determines whether a resource is seen as waste rather than a resource, based on the substance object's properties [67]. According to this definition, too many materials are classified as waste. As a result, it is forbidden to trade, mediate, transfer or receive the substance which is classified as waste, without registration or permit. These definitions prevent the innovative use of materials and textile residue and undermine sundry models of a circular economy [68].

In order to engage consumers, public policies need to be crafted so that they influence and shape consumer behavior. In a world where 85% of new clothes are thrown away within a year from the moment of purchase, over 144 million tonnes of textile waste are generated globally each year. (Since the 1960s there has been an 800% increase in the amount of textile waste buried) [5–7,69,70]. It is clear that innovative regulatory policies are

imperative to reducing textile waste. New laws must introduce a "polluter pays" dynamic at the individual and household level. In this context, a waste disposal fee (often called "pay as you throw" [71] as has been initiative by municipalities New Zealand [72,73], New Jersey and other jurisdictions.

In communities with pay-as-you-throw programs, residents are charged for the collection of municipal solid waste based on the amount they throw away. This contrasts with traditional policies where residents pay for waste collection through property taxes or a fixed fee, regardless of how much—or how little—trash they actually generate. The policy creates a direct economic incentive to create less waste [71], positively shaping consumers' clothing disposal patterns and presumably purchasing patterns as well.

### 3.1.2. Legal Liability for Violation of Garment Workers' Rights

One of the main reasons for the widespread environmental damage associated with the fashion industry, as well as the violation of workers' rights, is the practice of outsourcing, which allows manufacturers to avert legal and substantive liability. The production model of fast fashion companies emerged in the 1980s. An integral part of the development of the profit model for fast fashion companies during those years, was the radical change in marketing strategy. Accordingly, the number of collections per year increased from 2 or 4 to 52 collections today [3].

The conceptual basis of the success of fast fashion corporations rests on the production of an image of the product and its customers, as opposed to emphasizing production processes in general and high quality products in particular. According to this concept, the actual production of goods is almost a negligible part of the total activity of the corporation. It can therefore be outsourced and subcontracted [74]. Today, from both a legal and substantive perspective, fashion corporations should not be seen as companies for the production of clothing, but rather for the creation of image and branding.

Indeed, in recent decades, more and more fashion companies have relocated clothing production to developing countries, where they enjoy cheap labor, extensive tax breaks and lax environmental laws and regulations. Today, 97.5% of clothing sold in the US is imported [75]. The production of clothing in developing countries relies on cheap labor and has enabled the dramatic reduction in clothing prices. This directly resulted in the formation of the fast fashion model, notwithstanding its many environmental and social consequences [76].

At present, the fashion industry is largely based on branding methods: the major companies actually market their name but do little to produce the clothes that carry it. In fact, many companies refuse to disclose the names and addresses of their manufacturers' plants, under the pretext of trade secrecy and infringement of competition [77]. In the spirit of the sb 62 law, imposing legal liability for workers' conditions, will enable the protection of workers' rights and the reduction of environmental damage in the fashion industry, as well as improve the trackability of supply chains.

### 3.1.3. Advertising Standards for the Apparel Industry

Demand for fast fashion is driven by an aggressive advertising industry. An effective strategy for promoting sustainable fashion requires that countries address the more pernicious aspects of the associated marketing. In this context, on 1 January 2013, a unique law came into force in Israel: the 'Weight Restriction Law in the Modeling Industry', also known locally as the "Photoshop Law". The purpose of the law is to limit the exposure of the public in Israel to advertising that presents body images of extreme thinness, in order to prevent the development of eating disorders. The law stipulates, inter alia, that an advertiser must clarify in a commercial advertisement the use of editing for the purpose of narrowing body circumference if a commercial advertisement uses this technology. The clarification must appear in a prominent place in advertisements using a clear color and size, over an area no less than 7% of the total advertising area of the advertisement [78].

Such regulatory interventions in advertising are common in the area of cigarettes and tobacco products. Evidence of the causal role of marketing in the tobacco epidemic and the advent of the WHO Framework Convention on Tobacco Control have inspired more than half the countries in the world to ban some forms of tobacco marketing [79]. By 2009, more than 100 countries had banned some form of tobacco marketing [80]. Singapore, the first country to restrict tobacco advertising, enacted a law that limited retail displays to one pack front per brand variation. Australia adopted plain packaging legislation [81,82]. Applying this model, which restricts the publication and/or requires clarification of risks or production shortcomings, to "fast fashion", would contribute to consumer and producer awareness about its environmental and social implications.

Similar to Israel's 'Photoshop Law', which is designed to curtail the negative, external consequences of the fashion industry, requiring clothing ads to contain a clarifying sentence about the carbon footprint of the garments and employment conditions of workers in textile factories have the potential change societal perceptions and norms. Beyond environmental criteria, these clarifications could also report whether, or not employee conditions are monitored and if so, on whose behalf. Such an approach largely corresponds with the United Nations Environment Program which stated at COP26 that it is working on a series of guidelines to publish in the field of fashion, which will be released next year, with the purpose of preventing Greenwash [14].

It should be noted that green wash constitutes a widespread problem in the fashion industry, greatly affecting the willingness of consumers to consume sustainable fashion. For example, a new report by Changing Markets, entitled *Synthetics Anonymous: Fashion brands' addiction to fossil fuels* analyzed almost 50 major fashion brands. The study found that brands routinely deceive consumers with false "green" claims. Of the many brands that made sustainability claims, some 59% of green claims flouted the UK Competition and Markets Authority guidelines in some way. Examples of brands misleading consumers include claims that synthetic products are recyclable when no such recycling technology exists, when brands were not specific about the amount of recycled content included in the product or where claims are made with no supporting evidence given for products being labelled as "sustainable" or "responsible", or [83].

Urska Trunk, Campaign Manager at Changing Markets writes: "While brands are quick to capitalize on consumer concern by using sustainability as a marketing ploy, the vast majority of such claims are all style and no substance. While they greenwash their clothing collections, they are simultaneously dragging their feet on embracing truly circular solutions, for example by not making the necessary investments to ensure a future in which clothes can be recycled back into clothes" [84].

*3.2. Educational Initiatives*

Educational Campaigns to Reduce Consumption

Fundamental problems within the fashion industry originate from the pace of production and consumption, which exacerbate environmental and societal crises [69]. Moreover, while in other areas, such as renewable energies and transportation, there is a plausible hope for progress due to technological developments, when it comes to textiles, technologically this appears to be highly unlikely. For example, in the field of textile recycling, composting produces methane, which contributes to greater greenhouse gas emissions and global warming [35]. Even the nutrient value of decomposed textiles to soil is low. In other words, the heart of the matter is not the way waste is treated, but the amount of textiles purchased and the speed at which textiles become waste. Dr. Lindsey Drylie Carey, Principal Investigator for the SFES project and Senior Lecturer in Marketing at GCU, summarized the dynamics: "Overconsumption is the big problem, not the way we manufacture our clothes or take care of the by-products of excess consumption" [38].

The purpose of a circular economic approach in fashion is to extend the duration of time that garments are used and to maintain the value of apparel products and materials as long as possible. Both of these goals are better achieved through education than through

technological developments or regulatory interventions. It is easy to argue that educational solutions be part of a strategy to reduce consumption. Programs delivered in formal and informal education systems should focus on creating a new awareness not only about the use of clothing but also their environmental and social implications. Educators need to introduce alternative practices for using clothing longer, maintaining them better and investing in a smaller wardrobe with less content [69]. Establishing such new norms can provide an alternative approach to the prevailing paradigm of high consumption/fast fashion [85].

Educational programs can be developed in collaboration with new NGOs which have begun to emerge to address this newly recognized social problem. Organizations, such as the *Sustainable Fashion Project*, which brings together universities and business entities from the fashion industry, to find practical solutions in the field of education and higher education; EU-funded *Sustainability Fashion Employability Skills* (SFES) project; and *Fashion Revolution*, have developed a variety of tools, suitable for a wide range of audiences including: "Fashion Ethics Trump Card Game", quizzes on globalization, workers' rights, supply chain transparency, material sourcing, global citizenship, sustainable development and ethical business practices [38,86,87] The organizations target diverse audiences, ranging from toddlers at the age of 6, through teens and up to adults and students. Their focus is educational, offering a range of programs in formal and informal settings such as writing a letter to the person who made my clothes (designed for elementary schools) and a richer program about sustainable consumerism designed for university students (under the hashtag Haulternative) [88]. Dr. Lindsey Drylie Carey, principal investigator of the SFES project, concludes: "Education is at the base of any change in society. Fashion education needs to embed sustainability in as many areas as it can".

However, designers themselves are capable of educating the public about the problematic aspects of fast fashion and the potential of circular production to produce an environmentally preferable product. Murzyn-Kupisz, and Hołuj [89] argue that "Fashion design education should be seen as a chance to make aspiring designers aware of the challenges and potential of design for sustainability and equip them with the knowledge and skills necessary to implement sustainable fashion approaches." As society increasingly lionizes talented fashion designers, they have the potential to influence the thinking and consumer patterns of their fans.

### 3.3. Incentives

3.3.1. Carbon Taxes on Clothing

Some 51% of tax revenues in the EU come from income taxes, while only 6% come from resource taxes [37]. Shifting taxes from work to services is expected to encourage circular, work-based and service-based business models, over traditional linear, product-based business models. This is because maintenance, repair and renewal of clothes are labor-intensive and resource-intensive. Accordingly, instead of applying a tax on labor, a carbon tax can be applied, which will immediately affect the extent of natural resource utilization and the pollution outcomes of production processes.

A carbon tax is an emission tax, imposed on the production chain, specifically focused on the greenhouse gas emissions released into the air, establishing a direct connection between the tax and the damage to the environment [90]. In the context of the fashion industry, a carbon tax should target the fashion corporations, responsible for greenhouse gas emissions in their production processes, rather than a tax applied directly on consumers, notwithstanding their carbon contribution.

Imposing a carbon tax within the borders of the European Union, or a federal tax in the United States on the carbon footprint of fashion corporations clothing manufacturing countries, would increase government revenues and provide an economic incentive for corporations to adopt more sustainable economic models and practices. Economic incentives could be established to encourage collaborations among fashion companies. These might include establishment and support for *industrial symbiosis areas*, and hubs to support

sustainable projects and designers. British professor of design, Nigel Cross' summarized the associated dynamics: "scientific problem solving is done by analysis, while designers problem solve through synthesis" [24]. Accordingly, a constructive approach to addressing the environmental problems created by the fashion industry should start with support for designers who can design meaningful solutions.

Industrial symbiosis (IS) is a sub-field of industrial ecology engaging "traditionally separate industries in a collective approach to competitive advantage involving physical exchange of materials, energy, water, and/or by-products" [91,92]. The adoption of IS can create economic benefits for companies, as well as environmental and social benefits for society (e.g., Jacobsen, 2006; Taddeo et al., 2017 [93,94]). IS is also considered to offer a key strategy in supporting a circular economy [92,95].

In the context of fashion, knowledge sharing and the shared use of innovative technologies and textile remnant, can greatly help reduce the environmental costs of production, as well as support the development of innovative and ground-breaking thinking in designing clothes for longer-than-usual life cycles. Government assistance for such ventures would constitute support with a double dividend: producing a greater product value beyond merely economic support and reducing the pernicious environmental impacts of the fashion industry.

### 3.3.2. Incentives and Investments on Favorable Terms to Businesses Offering Sustainable Fashion Services along with Reduction of Regulation and Taxation for Thrift Stores

As mentioned, the profit cycles of businesses in the circular economy are spread over a longer period-of-time than conventional production [32,34]. This is especially true for businesses based on rental, repair and second-hand sales. The period of time that elapses from the initial investment to its return is significantly greater than the immediate profits reaped by parallel, fast-fashion businesses. The relatively slow return on initial investment means that many ventures in the field of sustainable fashion find it difficult to raise seed money to get started and loans to expand growth [30].

In addition, as described above, VAT does not favor rent-purchase relationships. Nor do sales taxes provide a boost for second-hand clothes, or clothes made in a circular economy. In fact, for products that are not new, VAT is paid twice (or even more often) by users [30]. This situation not only undermines a competitive business plan for companies who are environmentally preferable than their conventional rivals. It also points to the significance of providing funding sources for initial venture support.

Government support for thrift stores or sustainable fashion ventures, therefore, should become an important new component in climate change mitigation and other environmental programs. It can take the form of initial incentives and loans on favorable terms for businesses promoting sustainable fashion, as well as changing the taxation system to differential taxation in the field of fashion. Under such a system, higher tax rates would be levied on less sustainable fashion products, creating a competitive advantage to sustainable fashion firms. This would enable many of them to grow and increase their profits, even without a primary investor.

### 3.4. Certification
#### Standardization or Certification of Sustainable Fashion

Another government strategy to promote sustainable fashion involves ensuring accessibility by the general public to a given item of clothing's environmental and social impact via establishing and publicizing certifiable standards, on clear criteria. In recent years, more and more consumers seek sustainable consumption alternatives. Research by IBM [96] and the giant corporation Unilever [97] both found that a third of consumers would prefer products that match their values, even if they have to pay more [96,97]. Indeed, over a third of consumers consider environmental considerations when shopping.

A study by the *Fashion Revolution* movement [98] based on 5000 fashion consumers, aged 16–75, in Europe's five largest markets, found that consumers want brands and

governments to ensure that clothing is produced responsibly and sustainably. A majority want to know more about the social and environmental impacts of clothing. Some 59% of respondents in the survey reported that they would like to know how their clothes are made. An additional 77% said they wanted the law to obligate fashion companies to respect the human rights of all involved in production; 75% wanted the law to require environmental protection at all stages of clothing creation; 68% wanted the law to oblige companies to provide information on the social impacts of companies; and 72% answered that they expect the law to require companies to report whether they pay workers who manufacture their products a fair wage.

The increased willingness of consumers to pay more for sustainable fashion, alongside the growing expectation by consumers that governments intervene and compel companies to report on their environmental and social performance is an extremely significant development. It indicates that a pressing need exists for legislatures to act and establish standards along with a validation body to oversee sustainable fashion. Presumably, a new wave of such statutes would allow consumers to approach the issue according to clear, transparent and replicable criteria.

Certification programs for sustainable fashion would increase public confidence by preventing the ineluctable greenwash attempts by opportunistic producers [84]. Designing criteria for sustainable fashion, which rank companies 'performance in relation to waste, workers' rights, use of water resources and toxic chemicals, transparency and more, would also make it easier for the players in the industry to establish quantifiable environmental targets. To use the words of Richard Dictus, president of clothing chain, *Amphora*: "We currently require fashion companies to set carbon emissions targets, but we do not yet have the methodology to measure this. If it is visible and measurable, we can change it" [38]. At the same time, certification will make it far easier for consumers to select sustainable fashion, allowing them to identify sustainable products and purchase them in lieu of fast fashion (for a summary of proposed alternatives, according to the stages of the product cycle in a circular economy, see Table 2).

## 4. Policy Recommendations Based on Application of Selected Criteria

Given the importance of making the right policy choices, alternatives should be analyzed systematically and analytically, based on defined criteria. We suggest four common and widely accepted principles for policy evaluation: effectiveness, equity, sustainability and political feasibility. We ranked the different categories of policy options described above according to these principles. Clearly local cultural, political and social conditions affect policy implementation. Yet, the survey helps to highlight those alternative strategies that appear to be the most promising.

## 5. Methods

The research was conducted in Israel, engaging local experts based on an instrument that referred to local conditions. The Israeli fashion market is dominated by imports and Israelis typically feel themselves to be part of European society, participating in European song contests, basketball leagues along with other cultural and economic trappings of the Continent. Nonetheless, there may be idiosyncratic aspects of local social forces and culture which limit the application of the findings in other countries.

As part of the study, policies were ranked on a scale of 1 to 5, where 1 represents the lowest score and 5 represents the highest score. The ranking of alternatives was done with the help of an expert panel composed of

- The management team of 'Dress Well'—the movement for the promotion of fair fashion in Israel. (This NGO is the only environmental organization in Israel that focuses on the field of sustainable fashion);
- Five owners of second-hand clothing stores;
- Three independent designers who produce clothes, self-defined as "slow fashion"
- Two fashion activists;

- The Legal Adviser of the Joint Organization, the largest humanitarian Jewish organization in the world, which operates a special department for encouraging equal opportunity in Israel's business sector;
- The CEO of the Heschel Center for Sustainability, a Tel Aviv-based think tank that works to promote the "vision of sustainability and a just and cohesive society".

The panel met twice in order to rank the competing policy options. At the beginning of the first session, after a presentation by the authors, participants were asked to rate the alternatives from one to five, with one being the lowest score and five the highest. Each expert was asked to rate the alternatives individually and separately from the other participants. An open discussion was then convened about the alternatives, involving the pros and cons of each approach. The second session opened with a re-introduction of the policy categories. An hour-long discussion reconsidered the different alternatives. Finally, the ranking of alternatives was made jointly with a consensus position reached.

Policies were assessed according to four criteria: The effectiveness criterion examines the extent to which the proposed alternatives meet the overarching policy objective, meaning how much the competing policies encourage the reduction of consumption of fast fashion and its replacement with slow fashion. The sustainability criterion examines the extent to which the proposed alternatives promote protection of the environment. The feasibility criterion examines the extent to which the proposed solution is feasible politically and economically in light of the anticipated objections of entrenched interests and the ability to mobilize support against them. This criterion also considers the power relations between the countries required to implement the policy option and its standing among key stakeholders.

The latter criterion was chosen because the global fashion industry is one of the most lucrative industries in the world, incorporating international trade relations on a large scale. As Edwin Keh, chief executive of the *Hong Kong Research Institute of Textiles and Apparel* put it: "If you see the warning signs, but you're not creating a business model, or a logistics model, you're making a half-baked solution that makes you feel good. This works well in a lab but has no real-world application" [99]. Therefore, however good it may be, an alternative without political and economic feasibility will not be able to produce the desired changes.

The equity criterion examines how fair the alternatives are for all parties involved. With regard to the fashion industry, it specifically examines distribution of the costs and benefits, such as the employment conditions of the workers in textile manufacturing plants. Finally, the compliance criterion examines the feasibility of the proposed alternative regarding the regulated community's willingness and ability to meet the standards and constraints imposed on it.

## 6. Results

The following offers a cursory summary of the findings emerging form the analysis:

Waste removal fees along with changing the definitions of waste laws in the field of fashion and textiles: This alternative is ranked as having medium to low effectiveness (2). It holds the advantage of allowing many fashion designers and corporations to utilize textile residues more easily, while providing legal support for sustainable production strategies. In the absence of additional incentives, there are no guarantees that clothes manufacturers will choose to integrate textile residues during production. With regard to equity—the alternative is fair to all parties and does not discriminate against a given group of producers. On the other hand, it is unlikely that removal fees will improve the situation of production workers in textile factories at all, contributing little to general equity. Hence, for these criteria, it was given a ranking of (3). When considering the sustainability criterion this alternative has the potential to significantly reduce waste levels, giving it a high sustainability grade (4). However, it is important to remember that in the absence of an economic incentive for fashion corporations to use textile residues and stop considering them as waste, realizing this policy's sustainability potential will only be partial. Finally, regarding the criterion of political feasibility, few objections to changing

waste regulations are anticipated making the alternative particularly palatable for politically minded decision makers (5).

Waste disposal fees on consumers (pay as you throw): This alternative is expected to be highly effective. Encouraging consumers to see the disposal of clothes as part of the cost of the item itself can help change consumption patterns. Experience gained in localities in the U.S., New Zealand and other countries offer valuable lessons in designing most effective format for policy implementation (4). As for the equity criterion, avoiding additional surcharges for waste disposal does not harm anyone. All citizens enjoy the option of producing less textile waste and saving money. In fact, this policy alternative is particularly fair for groups and individuals who produce little textile waste and yet suffer from the negative externalities of increasing textile waste (4). With regard to the sustainability criterion, this alternative is a sustainable alternative, which is expected to reduce the quantity of waste, providing a clear incentive for consumers to change consumption patterns (5). Finally, applying the criterion of political feasibility reveals a relatively weak point for the waste fee alternative. Depending on local social and political dynamics, it may be strongly opposed by voters, who would perceive the policy as a new, highly conspicuous tax. This may lead many decision-makers at the political level to avoid adoption (2).

Applying legal liability to corporations for violations of workers' rights and environmental standards: This alternative is expected to allow corporations to be sued or fined for the negative externalities of the fashion industry. As a result, it may have the result of pushing corporate leadership to dramatically transform internal practices and adopt state-of-the-art waste reduction technologies. This alternative was deemed to be particularly effective (5). As for the equity criterion, the alternative is expected to contribute to protecting basic rights of millions of workers around the world, by making corporations legally accountable for industry practices. Therefore, the alternative is deemed to be particularly fair (5).

With respect to the sustainability criterion, this alternative can be expected to improve the environmental performance of companies. Yet, its main impact may be manifested in workers' rights because its scope is significantly broader than conventional environmental legislation (3). As for political feasibility, the realization of the alternative is expected to engender widespread opposition from influential economic stakeholders facing liability for worker conditions in their textile factories as well as for high environmental exposures. Because of the anticipated "pushback", the option is ranked as having a particularly low political feasibility (1).

Clarification requirements in advertising: Studies in the field of behavioral economics reveal that the provision of information may actually be the least effective policy tool for changing behavior [100–102]. In addition, mandating notification about the environmental consequences of the fashion industry is not necessarily expected to affect consumer considerations. The degree of effectiveness of this alternative is therefore questionable (2). As for the equity criterion, this alternative does not significantly harm any given party, but is equally unlikely to significantly benefit production workers. It is fair in the narrow sense, but not in a broader sense (3).

As for the sustainability criterion, based on the experience of the aforementioned "Photoshop Law," this alternative may improve the environmental performance of fashion corporations to a moderate extent. Yet, over time, the influence will decline. The option therefore receives only a moderate score (3). Finally, with regard to the criterion of political feasibility, there have innumerable calls in recent years to increase transparency in fashion corporation [83], along with statements of many fast fashion corporations, who promise to become "greener" [11,14,99]. It can therefore be assumed that such initiatives will not meet significant resistance and are relatively feasible (5).

Educational campaigns to reduce consumption: In contrast to merely providing short-term information, whose effectiveness is limited, a sustained educational process is considered to be a far more promising strategy. The criterion of effectiveness, therefore, receives a high score (4). It should be borne in mind, however, that the effects of any educational alternative will only be felt after a relatively long time. As for equity, this alternative is fair

to all parties, as it does not harm a given party and even encourages target audiences to purchase sustainable fashion from businesses in the field (5). With regard to sustainability, if conducted professionally with sufficient resources, educational campaigns constitute a sustainable alternative, which may even produce ripples of sustainable consumption in areas other than fashion (5). As for political feasibility, the alternative is not expected to raise resistance and is therefore considered to be especially practical (5).

Carbon tax in the field of fashion: The degree of effectiveness of a carbon tax depends on many variables, including the ability to accurately quantify the associated greenhouse gases and assign them to different products so that taxes truly reflect the amount of carbon emitted in production and delivery. Given the multiplicity of stages in the supply chain, which are often spread over several continents, and the lack of trackability in the fashion industry, it can be assumed that a carbon tax will be only partially effective (3). As for equity, this alternative is particularly fair, because it is essentially aimed at correcting market failures. Presumably, firms who cause unnecessary external environmental costs will be the only ones who pay the tax. It is also expected to increase trackability in the fashion industry, which in turn will contribute to the area of workers' rights in textile factories. At the same time, this alternative does not directly address employment conditions of workers (3).

As for sustainability, this alternative directly addresses the environmental implications of the fashion industry and is certainly expected to reduce externalities (5). Regarding political feasibility, past experience with carbon taxes in other areas [103] shows that its implementation encounters a myriad of political and even cultural pushbacks. In addition, the difficulty of accurately estimating carbon emissions may significantly complicate the application of the carbon tax. Nonetheless, given the mounting concern worldwide about the climate crisis, with sufficient political will, a carbon tax remains a practical and feasible policy (3).

Economic incentives for collaboration between fashion companies: This option includes the establishment and support of industrial symbiosis areas, and technology hubs that provide a home to sustainable projects and designers. Collaboration between companies is an essential component for promotion of sustainable fashion [14,24,104]. The higher the level of collaboration, the more effective developing sustainable fashion products are likely to be. In addition, past field experience in a number of areas [92–95] shows that the growth of ventures in industrial symbiosis sites leads to highly improved effectiveness. If a critical mass of talented professionals can be brought together, this alternative is projected to be highly effective (5).

As for equity, a "collaboration" alternative does not harm anyone but serves to strengthen positive models. It can therefore be assessed as particularly fair for business owners and entrepreneurs in the field of sustainable fashion. At the same time, where worker conditions in textile factories are concerned, the question arises as to whether, without penalties and sanctions, present exploitative transactions will end. This question is even more salient because most lucrative employment in the field takes place in large fashion corporations, while this alternative is primarily intended for start-ups and small businesses. In the absence of a clear answer to this question, this alternative receives only a modest score in the equity criterion (3).

This alternative, however, has been found to be particularly sustainable in a number of areas, significantly increasing the level of environmental performance (5). Regarding *political feasibility*, since the alternative does not impose negative sanctions on any given political interest group, it is not likely to engender any opposition (5).

Providing initial incentives and investments on favorable terms to businesses that offer services and reduction of regulation and taxation for thrift stores: Providing initial incentives in and of itself does not guarantee business success. Nonetheless, reducing regulation and taxation will certainly contribute to the increased prosperity of existing businesses in the field of sustainable fashion. Therefore, the alternative is deemed to be largely effective (4). As for equity this alternative corrects a distortion in taxation policy, which leads to unfair competition between established businesses or between second-hand shops and fast fashion brands. The alternative is considered to be fair and equitable. It

should kept in mind, however, that the provision of economic incentives to businesses is an economic policy whose financing may come at the expense of resources to support other public policies (4).

As for sustainability, this alternative increases the chances of success in creating sustainable businesses, helping them to successfully compete with businesses selling unsustainable products. It constitutes a highly sustainable initiative. It does not, however, encourage reduction of consumption, which should be a sustainable fashion programs' ultimate goal. Rather, it only replaces rapid fashion consumption with sustainable fashion consumption, which might be considered an intermediate goal (4). With respect to political feasibility, the alternative is relatively short-lived and is not expected to draw opposition from any of the parties involved. It is deemed therefore to be largely practical (5).

Standardization or certification of sustainable fashion, based on agreed criteria and accessibility of the standard to the general public: Standardization and signaling sustainable fashion for the public is not sufficient to replace fast fashion with sustainable fashion. However, certification and communication remain at the heart of a strategy to promote sustainable fashion when using a range of policy tools. It constitutes a preliminary and necessary step, "along the way", which should help promote a broader array of sustainable fashion regulations (4). Regarding the equity criterion, as the alternative, would be based on professional, clearly defined and uniform criteria, affecting all those involved in the field of fashion equally, it is expected to significantly increase equity and give a competitive-commercial advantage to businesses in the field of sustainable and fair fashion (5).

Similarly, with regard to the sustainability criterion, the alternative is expected to improve the environmental performance in the fashion industry. That is because clearly defining the qualities of sustainable clothing serves to mark a "baseline" and provide a competitive-commercial advantage to sustainable businesses in the field of fashion. These are important normative measures for iterative improvement in the industry (5). The weak point of this alternative is its limited political feasibility. It may be possible to implement certification processes, but it will certainly be very complex. Definitions for sustainable fashion, as mentioned, are not uniform and vary both in the sustainability and in the fashion community. A standardization process for sustainable fashion, or even an agreement among experts who would perform the process promises to be highly complex (2) and produce controversy.

Table 3 summarizes the results of the ranking exercise. Some alternatives appear to enjoy high rankings across all criteria (e.g., educational campaigns or economic incentives for collaboration). However, the analysis also suggests that in considering policy alternatives, decisions should be cognizant of associated trade-offs between competing criteria. A policy which may be extremely effective in improving sustainability may also be unfair or lack political feasibility.

**Table 3.** A summary table of the alternatives indicated according to criteria.

|  | Effectiveness | Equity | Sustainability | Political Feasibility | |
|---|---|---|---|---|---|
| Changing definitions of waste laws in the field of fashion and textile | 2 | 3 | 4 | 5 | 14 |
| Waste removal fee | 4 | 4 | 5 | 2 | 15 |
| Applying legal liability to corporations in relation to workers' rights and environmental performance | 5 | 5 | 3 | 1 | 14 |
| Clarification requirement in advertising | 2 | 3 | 3 | 4 | 12 |
| Educational campaigns to reduce consumption | 4 | 5 | 5 | 5 | 19 |
| Carbon tax on fashion products | 3 | 3 | 5 | 3 | 14 |
| Economic incentives for collaboration in sustainable fashion | 5 | 3 | 5 | 5 | 18 |

**Table 3.** *Cont.*

|  | Effectiveness | Equity | Sustainability | Political Feasibility |  |
| --- | :---: | :---: | :---: | :---: | :---: |
| Incentives and investments on favorable terms to businesses that offer services and reduction of regulation and taxation for thrift stores | 4 | 4 | 4 | 5 | 17 |
| Standardization/certification for sustainable fashion, based on agreed criteria and accessibility of the standard to the general public | 4 | 5 | 5 | 2 | 16 |

## 7. Discussion and Conclusions

During the past decade, the field of fashion has gained a more respectable intellectual and political status than it has enjoyed in the past. Inherent qualities of the fashion industry, which in the past deemed it seemingly unwieldly and an inappropriate target for sustainability regulation (e.g., its ephemeral temporality, incessant changing and being the "most accessible art to the masses") are now turned on their heads, empowering sustainable fashion advocates. A window has opened for transforming the dominant culture of fast, unsustainable fashion.

Only recently has the fashion industry been recognized as one of the most polluting and exploitative industries in the world. It produces extensive pollution at every stage of production, from growing and dyeing cotton, through global transportation, which increases air pollution, to landfilling at the end of use. It is based on cheap manpower, primarily in third world countries, and long-term employment without basic safety conditions, where workers are often subjected to physical or verbal abuse.

The connection between fashion and sustainability, however, is not limited to the contribution of the fashion industry to the climate crisis and the widening of social gaps across the world, far from it. Indeed, just as it today is part of the problem, fashion could be part of the solution. Fashion is one of the most important and influential factors in culture. It shapes the consciousness of many, implicitly dictating what is "right" and what is not and contains within it opportunities for endless creativity and ground-breaking thinking. It can not only be part of solving the climate crisis, but also help raise awareness about sustainability challenges in general.

In line with this view, in 2019 the United Nations launched The United Nations Alliance for Sustainable Fashion, with the aim of contributing to Sustainable Development Goals (SDGs) through coordinated action in the field of fashion. The negative environmental and social impacts of the fashion industry, make it essential and a potential tool in attaining sustainable development goals [31].

The value of this report and the several policy proposals it contains goes far beyond the boundaries of merely improving the fashion industry. As the vast majority of people in the world wear clothes, environmental policy in the field of fashion can affect the thinking of countless individuals around the world. Well-considered sustainable fashion initiatives and their implementation have the power to significantly help engage the public in an effort to address the climate crisis and environmental injustice.

Promoting such a complex and important issue as sustainable fashion requires an integrated public policy, which combines several alternatives. In our view, a coherent policy must start by standardizing the field of sustainable fashion and setting uniform and agreed-upon criteria for what constitutes sustainable fashion. While this is a particularly complex task in practice, it is a prerequisite for effective public policies designed to promote sustainable fashion. In other words, it is not possible to reach a policy objective without clearly defining where the industry needs to go and quantifying associated goals.

From a long-term perspective, action must be taken to promote educational programs in the field of sustainable fashion. As mentioned, a well-designed campaign can be very effective, but its consequences are expected to be felt after a relatively long period of time

and therefore should be started as soon as possible. It also constitutes a "no regret" policy, which has few if any risks and can be initiated in parallel to a standardization process.

As a second step, the ecosystem for businesses in the field of sustainable fashion must be improved in order to support collaboration among businesses in the field of sustainable fashion. At the same time, it is important to increase available fashion alternatives that are accessible to the general-public (geographically and economically). To this end, economic incentives for collaboration between fashion companies, and providing initial incentives and investments on favorable terms to businesses with sustainable fashion products are important measures. Reducing regulation and taxation for thrift stores should be pursued alongside changing the broader definitions of waste in laws regulating the field of fashion and textiles.

Finally, the most difficult aspect of a sustainable policy, in both practical and political terms, involves applying legal liability to corporations in relation to workers' rights and environmental performance. Although such a policy receives low ranking in terms of political feasibility, this in no way diminishes its importance. Establishing liability for the abuse of workers in the fashion industry remains the most effective intervention to increase equity and combat social injustice. Elected officials tend to be attentive to public opinion. With sufficient attention and public outcry, the political calculus for this alternative is subject to change.

Past experience shows that even the heads of huge fashion corporations tend to be attentive to public opinion. Often their reactions are quicker than those of elected officials. This was the case with the "Who made my clothes" campaign in 2014 and with the "PayUp" campaign in 2021. The "Who made my clothes" campaign arose in response to the Rana Plaza disaster in 2013 in which 1147 workers were killed and another 3000 were injured. On the anniversary of the disaster, the Fashion Revolution movement called on bloggers from around the world to upload a photo to Instagram with their clothing label with the hashtag "Who made my clothes" while tagging fashion companies. The fashion corporations responded to the campaign quickly, uploading photos of the farmers and production workers to Instagram, along with the hashtag "I made your clothes", offering information about their terms of employment. The disclosure of the information led to a certain improvement in the conditions of workers as well as to an increase in transparency and the ability to monitor the production process of the clothes in the supply chains [52,105].

The "PayUp" campaign was created in response to the refusal of fashion corporations to pay textile factories for goods. When the COVID-19 epidemic hit Europe and the US for the first time, fashion corporations responded by transferring economic risk to the edge of supply chains, meaning to factories. They canceled all pre-crisis orders, forcing factories to incur 40 billion dollars in losses. To this end, activists from around the world launched a "PayUp" campaign calling on corporations, through social media, to pay the debt [11,53]. In light of the public outcry, many corporations have pledged to pay for orders, and as of publication (November 2021), USD 27 billion has returned to factories.

Regulation, legislation and public policy should continue to play increasingly key roles in promoting sustainable fashion. We show that there can be trade-offs associated with different sustainable policies, where effectiveness may come at the expense of equity, or the most sustainable alternative may be politically unfeasible. Yet, consumers must remember that the fashion industry, more than any other polluting sector, depends on its customers for its continued operation. When it comes to consumer choices, every consumer is a decision maker. Consumer responsibility that encourages people to ask relevant questions and choose slow over fast fashion can lead to a significant change in the fashion industry, a change which will be no less powerful and even more meaningful than top-down changes led by decision makers.

**Author Contributions:** Conceptualization, M.P.M. and A.T.; Methodology, M.P.M.; Formal Analysis, M.P.M.; Investigation, M.P.M.; Writing—Original Draft Preparation, M.P.M.; Writing—Review and Editing, A.T.; Visualization, M.P.M.; Supervision, A.T. All authors have read and agreed to the published version of the manuscript.

**Funding:** This research received no external funding.

**Institutional Review Board Statement:** Not applicable.

**Informed Consent Statement:** Not applicable.

**Conflicts of Interest:** The authors declare no conflict of interest.

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
