# Peer review of "Regulation for Promoting Sustainable, Fair and Circular Fashion"

_sustainability, doi:10.3390/su14010502_

Round 1

Reviewer 1 Report

The topic is interesting but very easy to confuse between so many subtopic related to regulation in circular fashion. Introduction is too long. I suggest insert more main titles (1, 2, 3 etc.) to clarify each subtopic.
I will grateful if you present a schema showing the relations between all constructs used. In page 3, I humble suggest visualize policies, criterions and main highlaigths (and references maybe)- table/schema.
The redaction of the paper is a little confusing to follow, in my opinion.

Author Response

Reviewer 1:

Comment: The topic is interesting but very easy to confuse between so many subtopic related to regulation in circular fashion. Introduction is too long (I suggest insert more main titles (1, 2, 3 etc.) to clarify each subtopic.

Response: It is always challenging when two reviewers provide recommendations which are in conflict of each other. Reviewer 2 called for us to supplement the opening section rather than shorten it. (About half of his/her requests for supplemental additions dealt with the opening section.) So in order not to dismiss Reviewer 1’s comment that the introduction be shortened entirely, we have tried to only make the most critical additions in light of Reviewer 2’s excellent suggestions – and respectfully request that we leave the revised (but only modestly shortened) introductory opening sections as they are in the revised MS.

Comment: I will grateful if you present a schema showing the relations between all constructs used. In page 3, I humble suggest visualize policies, criterions and main highlaigths (and references maybe)- table/schema.
The redaction of the paper is a little confusing to follow, in my opinion.

Response: We have now included three visually convenient presentation and several more main titles. We believe that this meets the expectations of Reviewer 1.

Reviewer 2 Report

Great work by the authors, it needs some work to strengthen the paper. Please see below. The reviewer did not review the content after Methods section as that requires a major revision. 

Tittle: Are the three terms sustainable, fair and circular different? If the paper is on just circular- then title should not be misleading

Line 15-16 mentions four categories but authors list three categories. Perhaps add a comma after incentives as Certification is the fourth category.

Line 37-38 – It’s the cost or price of the same product, product itself is not different. Please make it clear

Line 39-40- how is it incentivizing consumer to increate their transportation, please explain.

Line 40-41- gives a different factor than 39-40.

Lines 35-42- please read carefully and see if it is communicating what the authors intended to say. To the reviewer, it is confusing

Line 49-Please be consistent in the terminology. Is economic policy same as  tax policy that is stated in previous lines?

Line 49-55- any reference document/citation of the policy document

Line 89-89- fashion industry second most exploitive in world…which is the first one, it is important to mention it as the reader may not know.

Line 112-113- the criteria listed in abstract are not aligned accurately in the article

Line 143-144- source on how circular economy proposes an alternative for social justice?

Line 152- a new idea.. what is Wise consumption and how it relates to the paper?

Line 165- I think authors want to say reduced utilization of virgin materials? Please clarify

Line 169-175 -the definition should be discussed earlier in the paper and authors should establish the operational definitions of the terms they are using

Line 219- source of the quote

Line 228-229- the interpretation of the reference #32 is not accurate.

Line 232-239- uses the term technical that should be replaced by another term as technical in circular economy model refers to technical cycle so it gets confusing

Line 258- why the expanded form of VAT is referred now- it should have appeared sooner in the paper. Or authors stay consistent by using just VAT

Line 314-315- clarify that consumers concern is for products made from recycled materials?

Line 323 MUD should be referred as an example..

Line 334-339- repetition from previous section

Line 361- rephrase

Line 370-371 what

Line 403-405 counters the previous statement from line 143-144

Line 430- are these regulations or initiatives or a mix of both? Clarify in the paper

Line 571- a very different perspective- please give context to make it fit in the paper- it is an imp. Aspect

Line 599-609 should be presented in the same structure as previous paras and focus on the regulation

Line 616- education initiatives need further development

Line 746- cost of certification should be discussed and audits and their ethics should be incorporated

3.1 Methods: It should clear state to region and how were they selected. The examples in the earlier part of paper are all from EU and US, but the methodology uses experts from a different market? I would reread the paper after Methods section has been revised. As there is a geographical and political limitation to this work that should be duly stated.

Author Response

Reviewer 2:

Comment: Great work by the authors, it needs some work to strengthen the paper. Please see below. The reviewer did not review the content after Methods section as that requires a major revision. 

Comment: Tittle: Are the three terms sustainable, fair and circular different? If the paper is on just circular- then title should not be misleading

Response: the paper includes an analysis assessing, inter alia, the equity and sustainability of different policies.   We believe that the present title accurately represents the content of the article.

In addition, in the article, in lines 161-181 and in lines 435-440, we distinguish between circular fashion and fair and sustainable fashion, and emphasize the complexity of the definition of sustainable fashion.

We argue (in the 96-113 lines) that circular fashion is important, but it is not enough to solve all the inherent problems of the fast fashion production model, including issues related to workers’ rights and consumers' wasteful consumption patterns. Therefore, a broader reference to circular fashion is required, that is, a reference to fair and sustainable fashion as well.

Comment: Line 15-16 mentions four categories but authors list three categories. Perhaps add a comma after incentives as Certification is the fourth category.

Response:

Our article refers to four different types of

  • Command and Control Interventions,
  • Educational Initiatives,
  • Incentives and
  • Certification.

Hopefully this will address Reviewer 2’s concerns.

Comment: Line 37-38 – It’s the cost or price of the same product, product itself is not different. Please make it clear;

Response: Reviewer 2 is correct. The problematic sentence now reads:

“The exact same product sold at a local and foreign site would be priced differently, with the foreign site less expensive - solely due to the tax gap.”

Comment: Line 39-40- how is it incentivizing consumer to increase their transportation, please explain.

Response: We agree.  Sentence now reads more clearly:

“In other words, the tax system incentivizes consumers to purchase internationally, which increases transportation expenditures and associated environmental impacts.”

Comment: Line 40-41- gives a different factor than 39-40.

Response: Sentence now reads: “Elimination of the exemption reduces the possibility of VAT fraud and increases VAT revenues for EU member states, by as much as 7 billion euros [2].

We believe the sentence is now more coherent.

Comment: Lines 35-42- please read carefully and see if it is communicating what the authors intended to say. To the reviewer, it is confusing

We the two changes made in lines 35-42, we believe that the passage now is clear.

Comment: Line 49-Please be consistent in the terminology. Is economic policy same as  tax policy that is stated in previous lines?

Response: In response to this concern, the sentence has been changed from “economic policy” to “taxation policy”. We believe that this should address the Reviewer’s concerns.

Comment: Line 49-55- any reference document/citatioan of the policy document

Response: Lines 49-55 are based on sources number 1 and 2 in addition to drawing our own conclusions. Therefore, we formulated thjis paragraph carefully and wrote "This new taxation policy can potentially" and "which may as a result, lead to a reduction in overall consumption". In addition we have added a reference to the relevant sources as requested.

Comment: Line 89-89- fashion industry second most exploitive in world…which is the first one, it is important to mention it as the reader may not know.

Response: Revised sentence includes mention the cell phone industry as the most exploitative industry. Sentence reads: "These figures suggest that in addition to being the second most polluting industry, the fashion industry is also the second most exploitative one in the world (Second only to the cell phone industry)."

In order to be on the safe side, we also added basic clarification regarding being the second most polluting fashion industry. Revised sentence includes mention of “energy” as the most polluting: Sentence reads:

“The associated environmental benefits are all highly germane to the fashion industry, which is now considered to be the second most polluting industry in the world (energy retains the dubious honor of top polluter)…..”

Comment: Line 112-113- the criteria listed in abstract are not aligned accurately in the article.

Response: Thanks for catching that. The criteria appearing in the abstract are now in the same order as those in the article.

Comment: Line 143-144- source on how circular economy proposes an alternative for social justice?

Response: This claim is based on source 17 and source 18, we have added this in the body of the article. It is important to mention that the claim that a circular economy supports social justice is not acceptable to all scholars and is controversial. There are scholars such as Hervé Corvellec et al, who in their article "Introduction to the Special Issue on the Contested Realities of the Circular Economy" (https://www.tandfonline.com/doi/full/10.1080/14759551.2020.1717733 ) argue that the way of applying a circular economy does not necessarily support social justice. In our opinion, the article is comprehensive enough and includes our position that a circular economy is not enough to properly address the employment conditions of workers in textile factories, but if necessary we can easily add the views of additional researchers.

Comment: Line 152- a new idea.. what is Wise consumption and how it relates to the paper?

Response: We agree, the wording of the sentence was not clear enough. We have reformulated it and it now appears as follows: " Wise consumption, meaning minimizing unnecessary purchases, preferring products with circular characteristics and choosing cooperative models of consumption that prevent storage and disposal of finished products also contributes to the circular economy "

Comment: Line 165- I think authors want to say reduced utilization of virgin materials? Please clarify

Response: We have changed the sentence which now specifies “utilization of virgin raw materials”…

Comment: Line 169-175 -the definition should be discussed earlier in the paper and authors should establish the operational definitions of the terms they are using

Response: At present, the definition of sustainable fashion appears during an early, “introductory” section of the article. In addition, we have shortened the introductory chapter so that now the definition appears in an earlier place. We believe that this is still a reasonable place for the definition as it proceeds the substantive analysis of the article.  If editor is unconvinced, we could easily move this paragraph.

Comment: Line 219- source of the quote

Response: Thank you for the comment. We added the source of the quote (source number 14)

Comment: Line 228-229- the interpretation of the reference #32 is not accurate.

Response: The sentence has been rewritten with a direct quote from the UN position paper on the subject. After the revision, it is now accurate.

  In this context, it is worth mentioning that the UN has defined the fashion industry as being ““of high relevance to integrating the targets of the seventeen Sustainable Development goals”. [32].

Comment: Line 232-239- uses the term technical that should be replaced by another term as technical in circular economy model refers to technical cycle so it gets confusing.

Our use of the term “technical” is based on the term used in the literature (see: van der Laan, K. Circular Revenue Models.).  We believe that it does not make sense to create a different term than that which has been used in this context.

Comment: Line 258- why the expanded form of VAT is referred now- it should have appeared sooner in the paper. Or authors stay consistent by using just VAT

Response: We agree: The term “Value Added Tax” on lie 273 is now replaced with VAT – and the first mention of VAT on line 28 is not preceded with the acronym fully spelled out “Value Added Tax”.

Comment: Line 314-315- clarify that consumers concern is for products made from recycled materials?

Response: the sentence has been corrected to include mention of low consumer awareness of recycled materials.

Comment: Line 323 MUD should be referred as an example..

Response: The present text reads: “In just one of many examples, the economic barrier of VAT, which does not favor rent-purchase relationships, negatively affects the dynamics of MUD Jeans”.

We believe that this indeed refers to MUD jeans as an example.

Comment: Line 334-339- repetition from previous section

It can be argued that this paragraph simply provides an additional example of tax distortion. But at the recommendation of the Reviewer, we have erased it.

Comment: Line 361- rephrase

Response: The sentence has been rephrased as requested and now is grammatically preferable. New sentence reads: “Worldwide, 130 billion garments are produced annually, of which 80 billion are sold, yet, less than a single percent are recycled”

Comment: Line 370-371 what

Response: These two sentences have been merged into a more coherent single sentence: “It should be mentioned that the environmental impact of low-cost garments typically is not felt in the location where they are purchased but rather primarily affect the locations where they are produced, buried or burnt.”

Comment: Line 403-405 counters the previous statement from line 143-144

Response: Line 143-144 explains that circular economy theory: “proposes an alternative model that supports sustainable development, social justice and economic well-being.”

Line 403-405 – speaks of the lack of sufficient attention directed to social aspects in the circular economy methods used for the fashion industry. So that is not actually a contradiction. But the reviewer believes that this is not made clear enough and the point is well taken. So the sentence was rewritten as follows:

“Another lacuna in present circular economy methods involving the fashion industry is the narrow focus on the environmental aspects of production at the expense of adequate consideration of social aspects.”

We believe that this addresses the reviewer’s concern.

Comment: Line 430- are these regulations or initiatives or a mix of both? Clarify in the paper

Response: Interesting point. In fact it is a mix. So we have changed the sub-title to: “Existing initiatives that promote sustainable fashion” – as initiatives can include regulations – but the opposite is not true.

Comment: Line 571- a very different perspective- please give context to make it fit in the paper- it is an imp. Aspect

Response: An excellent suggestion. This paragraph now begins with three transition sentences to avoid the abrupt start of a new topic:

“Demand for fast fashion is driven by an aggressive advertising industry. An effective strategy for promoting sustainable fashion requires that countries address the more pernicious aspects of the associated marketing. In this context, on January 1, 2013, a unique law came into force in Israel: the 'Weight Restriction Law in the Modeling Industry', also known locally as the “Photoshop Law".

Comment: Line 616- education initiatives need further development

Response: Education for sustainable fashion can be the topic for an entire article.  In response to the reviewer’s comment, we added an additional paragraph on this topic. But we feel that the article is already quite expansive at this stage and that the present general discussion is sufficient.

Comment: Line 746- cost of certification should be discussed and audits and their ethics should be incorporated

Response: This is an interesting topic but one that is entirely new and in our opinion beyond the scope of the present article which is already quite expansive. We would respectfully request that the editor leave this issue for a future research initiativer.

Comment: 3.1 Methods: It should clear state to region and how were they selected. The examples in the earlier part of paper are all from EU and US, but the methodology uses experts from a different market? I would reread the paper after Methods section has been revised. As there is a geographical and political limitation to this work that should be duly stated.

Response: This is a good point. We have added an opening paragraph putting this caveat forward. At the opening of the Methods section, the article currently reads:

“The research was conducted in Israel, engaging local experts based on an instrument that referred to local conditions.  The Israeli fashion market is dominated by imports and Israelis typically feel themselves to be part of European society, participating in song contests, basketball leagues along with other cultural and economic trappings of the Continent. Nonetheless, there may be idiosyncratic aspects of local social forces and culture which limit the application of the findings in other countries. “

Reviewer 3 Report

Mansuscipt highlights the link between  fashion and  sustainability, however, which is not limited to the contribution of the fashion industry to the climate crisis and the widening of social gaps around the world. In the review paper, a review of the literature was given, in a large thematic and date breadth, with an emphasis on contemporary research.
I am of the opinion that this work is a valuable contribution to the guidelines of technological and design achievements and collaboration.

It is obligatory to adapt the work to the instructions for authors, ie the form determined by the editor.

Author Response

Reviewer 3

Comment: Mansuscipt highlights the link between fashion and sustainability, however, which is not limited to the contribution of the fashion industry to the climate crisis and the widening of social gaps around the world. In the review paper, a review of the literature was given, in a large thematic and date breadth, with an emphasis on contemporary research.
I am of the opinion that this work is a valuable contribution to the guidelines of technological and design achievements and collaboration.

It is obligatory to adapt the work to the instructions for authors, ie the form determined by the editor.

Response: We believe that Reviewer 3 has offered a strong, positive review and are grateful for his/her position.  We believe that our present MS meets the instructions for authors and is ready for publication.

Round 2

Reviewer 1 Report

The paper has been modestly improved but enoght for publishing.

The main aim of the paper is clear, and the topic interesting. The size of table should be reviewed (I dont know if is possible not divide it).

Congrats

Reviewer 2 Report

Thank you for addressing the feedback. All the best.
